# A statistical framework for assessing pharmacological responses and biomarkers using uncertainty estimates

**Dennis Wang[1,2]\*, James Hensman[3], Ginte Kutkaite[4,5], Tzen S Toh[6], Ana Galhoz[4,5], GDSC Screening Team[7], Jonathan R Dry[8], Julio Saez-Rodriguez[9], Mathew J Garnett[7], Michael P Menden[4,5,10]\*, Frank Dondelinger[11]\***

[1]Sheffield Institute for Translational Neuroscience, University of Sheffield, Sheffield, United Kingdom; [2]Department of Computer Science, University of Sheffield, Sheffield, United Kingdom; [3]PROWLER.io, Cambridge, United Kingdom; [4]Institute of Computational Biology, Helmholtz Zentrum München—German Research Center for Environmental Health, Neuherberg, Germany; [5]Department of Biology, Ludwig-Maximilians University Munich, Martinsried, Germany; [6]The Medical School, University of Sheffield, Sheffield, United Kingdom; [7]Wellcome Sanger Institute, Cambridge, United Kingdom; [8]Research and Early Development, Oncology R&D, AstraZeneca, Boston, United States; [9]Institute of Computational Biomedicine, Faculty of Medicine,Heidelberg Universityand Heidelberg University Hospital, Bioquant, Heidelberg, Germany; [10]German Center for Diabetes Research (DZD e. V.), Neuherberg, Germany; [11]Centre for Health Informatics, Computation and Statistics, Lancaster Medical School, Lancaster University, Lancaster, United Kingdom

**\*For correspondence:**
dennis.wang@sheffield.ac.uk (DW);
michael.menden@helmholtz-muenchen.de (MPM);
fdondelinger.work@gmail.com (FD)

**Group author details:**
GDSC Screening Team See page 17

**Abstract** High-throughput testing of drugs across molecular-characterised cell lines can identify candidate treatments and discover biomarkers. However, the cells' response to a drug is typically quantified by a summary statistic from a best-fit dose-response curve, whilst neglecting the uncertainty of the curve fit and the potential variability in the raw readouts. Here, we model the experimental variance using Gaussian Processes, and subsequently, leverage uncertainty estimates to identify associated biomarkers with a new Bayesian framework. Applied to in vitro screening data on 265 compounds across 1074 cancer cell lines, our models identified 24 clinically established drug-response biomarkers, and provided evidence for six novel biomarkers by accounting for association with low uncertainty. We validated our uncertainty estimates with an additional drug screen of 26 drugs, 10 cell lines with 8 to 9 replicates. Our method is applicable to any dose-response data without replicates, and improves biomarker discovery for precision medicine.

## Introduction

The failure rate for new drugs entering clinical trials is in excess of 90%, with more than a quarter of drugs failing due to lack of efficacy (*Arrowsmith and Miller, 2013*; *Cook et al., 2014*). The rapid development of technologies for deep molecular characterisation of clinical samples holds the promise to uncover molecular biomarkers that stratify patients towards more efficacious drugs, a cornerstone of precision medicine. In oncology, we can identify potential biomarkers of drug response in high-throughput screens (HTS) of patient-derived cell lines; these biomarkers need to be then validated in patients.

Assessment of cell line drug response typically involves treatment with multiple concentrations of the compound, followed by measurement of the amount of viable cells after a fixed period of time for each dose, and derivation of a dose-response curve. The drug response is commonly then summarised by measurements taken from this curve, most often the concentration required to reduce cell viability by half that is $IC_{50}$, or the area under the curve that is AUC. Currently the two largest in vitro drug screening studies, the Genomics of Drug Sensitivity in Cancer (GDSC) (*Garnett et al., 2012*; *Iorio et al., 2016*) and the Cancer Therapeutics Response Portal (CTRP) *Rees et al., 2016* have shown that some clinically-actionable biomarkers of drug response can be concordantly discovered (*Iorio et al., 2016*; *Seashore-Ludlow et al., 2015*), and that different properties and mechanisms of drug response are best captured by different metrics dependent on the dose-response curve (*Fallahi-Sichani et al., 2013*).

Most HTS efforts focus on increasing throughput (*Iorio et al., 2016*; *Seashore-Ludlow et al., 2015*) and thereby often neglect experimental replicates, which renders it impossible to correct for experimental noise, resulting in uncertainty for the estimated drug-response metrics (e.g. $IC_{50}$ value). Extrapolating $IC_{50}$ values beyond the tested drug concentration range is particularly challenging and often unaccounted for in quality control metrics (*Haibe-Kains et al., 2013*; *Haverty et al., 2016*). Most published studies using machine learning algorithms or mechanistic models for predicting drug response and biomarkers assume that the measured drug responses are precise (*Costello et al., 2014*; *Keshava et al., 2019*; *Menden et al., 2019*; *Silverbush et al., 2017*). If this assumption is not met and there is high uncertainty in the measured drug-response values, the utility of these methods for enhancing drug development may be severely limited (*Costello et al., 2014*; *Menden et al., 2019*; *Silverbush et al., 2017*). Experimental noise can be reduced by adding experimental replicates, however, this either reduces the throughput of the screen or increases the cost. Most current models for curve fitting and describing dose-response data have primarily assumed that cell viability has a sigmoidal relationship to the logarithm of the dose concentrations of the drug (*Dawson et al., 2012*; *Wang et al., 2010*). Whilst some models are more flexible by allowing many inflection points in the dose-response curve (*Di Veroli et al., 2016*; *Vis et al., 2016*), their main output is a single drug-response value that does not fully capture the uncertainty in the measurements (*Fallahi-Sichani et al., 2013*).

Gaussian Processes (GP) are a flexible, probabilistic modelling technique that has been successfully used to measure uncertainty in noisy gene expression datasets (*Lopez-Lopera and Alvarez, 2019*) and has been incorporated into machine learning prediction of cell fates (*Boukouvalas et al., 2018*). This technique has been shown to cope well with regression tasks on dependent data and high dimensional covariates (*Rasmussen and Williams, 2005*; *Shi and Choi, 2011*). Instead of fitting a single function to the data, GPs allow for a flexible range of beliefs about the function underlying the data (*Tian et al., 2017*). In the case of cell line drug responses, this can be conceptualised as fitting a range of curves that have equivalently strong fit to the data. We can sample from the inferred posterior distribution over functions, that is the variance between these curves, to generate uncertainty estimates of quantities of interest, in our case, properties of the dose-response such as $IC_{50}$.

GPs have been recently utilised to identify and guide experimental validation of compounds, on top of being applied to protein engineering and imputing gene expression values (*Hie et al., 2020*). GPs have also been used in conjunction with neural networks to model dose-response curves as a function of molecular markers (*Tansey et al., 2018*). The main objective in this work was to predict drug response using the molecular measurements, and the non-linear nature of the prediction model makes interpretation for the purpose of biomarker detection challenging. By contrast, we aimed to develop a model that could provide interpretable summary statistics with uncertainty estimates that can be flexibly used to improve biomarker detection.

In this study, we therefore introduce a new GP regression approach for describing dose-response relationships in cancer cell lines that quantifies the uncertainty of the model fitted to measured responses for each single experiment, and we show that estimates of $IC_{50}$ values within the tested concentration range correlates with confidence intervals obtained experimentally from replicate experiments. Subsequently, we use our new dose-response model to identify genetic sensitivity and resistance biomarkers in standard statistical tests (e.g. ANOVA). We demonstrate how the flexibility of the GP dose-response modelling can be further exploited in a Bayesian framework to identify novel biomarkers. We also describe the variation in the level of drug response uncertainty across

cancer types and drug classes. By accounting for the uncertainty in dose-response experiments, detection of clinically-actionable biomarkers can be enhanced.

# Results

## A probabilistic framework for measuring dose-response and predicting biomarkers

We analysed in vitro screening data on 265 compounds across 1,074 cell lines (*Iorio et al., 2016*). In those experiments, we quantified the amount of cytotoxicity after four days of compound treatments at each dose compared to controls (*Figure 1A*). The relationship between the dose and response (decrease in cell viability) was first described using a dose-response curve derived with a sigmoidal

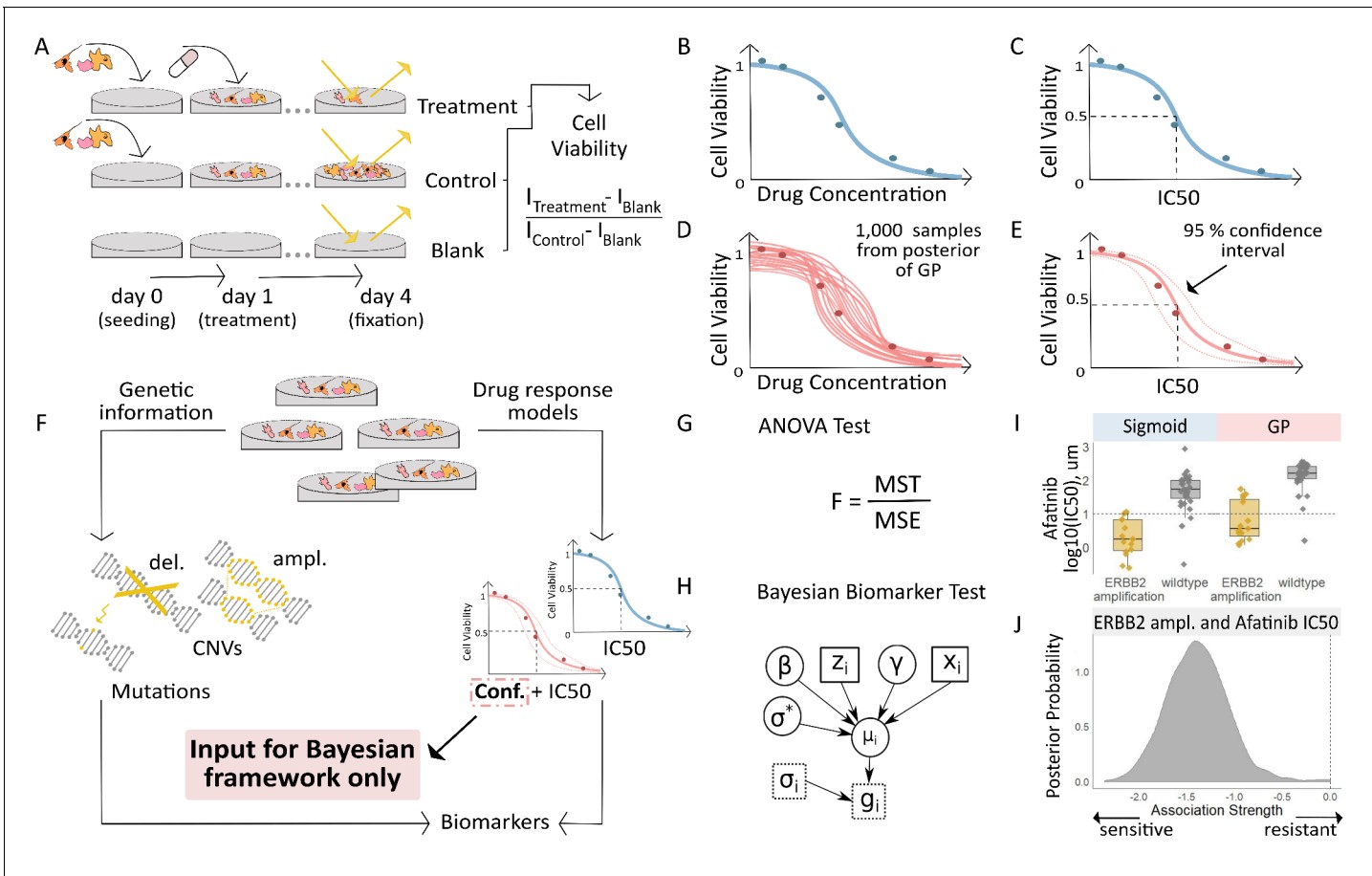

**Figure 1.** Workflow for fitting of Gaussian Process models to dose-response curves and estimating their uncertainty. (**A**) Large-scale drug screens test cell lines with different drugs and at different doses are used to obtain dose-response data. (**B**) Typically, for each drug tested in a cell line, the sigmoid model is fit to the drug-response data and (**C**) the overall measures of response ($IC_{50}$, AUC, etc.) are extracted. (**D**) For each drug tested in a cell line, we fit a GP model to the dose-response data. The GP allows us to sample from a distribution of possible dose-response curves, obtaining a measure of uncertainty. (**E**) From these curves, we can extract overall measures of response, such as $IC_{50}$, and importantly, their 95% confidence intervals. (**F**) Mutation markers for each cell line can be determined based on presence/absence of single nucleotide polymorphisms (SNPs) in key genes. Both the drug-response estimates and the mutation markers are used to compute (**G**) the F-statistic for ANOVA, and (**H**) Bayesian test for biomarker association. The drug-response summary measure $g_i$ for cell line i is modelled via a cell line- specific mean $\mu_i$ and standard error $\sigma_i$. The mean is defined as a linear effect $\beta$ of the biomarker status $z_i$ and a further effect $\gamma$ from any remaining covariates $x_i$, such as tissue type. The parameter $\sigma^*$ is the standard deviation of $\mu_i$. (**I**) Boxplots illustrate the differences in the estimated mean $IC_{50}$ of *ERBB2* amplified and non-amplified breast cancer cell lines treated with afatinib. An ANOVA test was used to test this difference in means but did not consider uncertainty in each $IC_{50}$ estimate. (**J**) We estimated posterior distributions of gene association using the Bayesian model, that is the effect of a genetic mutation on the $IC_{50}$ measurement of drug response. Distributions centred on zero indicate no effect whilst distributions on either side of zero indicate positive or negative effects of mutations on drug response.

function (*Figure 1B and C*). This assumes that the number of viable cells decreases at an exponential rate, then slows down and eventually plateaus at a lower limit. Since it was costly to test all possible doses, the sigmoid function was used to extrapolate the response at concentrations that had not been tested and to estimate overall measures of response, such as $IC_{50}$ or AUC values, for downstream analysis. However, considering that each experiment tested only between five and nine dosage concentrations per experiment in GDSC, and a maximum of 16 in CTRP, the tightness of fit of the dose-response curve to the data points and therefore the level of uncertainty about the inferred response may vary. We utilised the probabilistic nature of GP models to quantify the uncertainty in the dose-response experiments as an alternate approach (*Figure 1D*). We sampled from the fitted GP and used the posterior distribution to quantify the uncertainty in curve fits for each experiment. We again generated summary statistics, $IC_{50}$ and AUC values, by taking the average of the GP samples and also quantified the level of uncertainty for these statistics (*Figure 1E*). The GP model has the advantage that it models outliers at higher doses as one component of a two-component Beta mixture in the model (see Materials and methods). Such outliers are typically the result of an experimental failure, and cannot be modelled using simple Gaussian noise without over-estimating the noise parameter.

After fitting the dose-response data using the sigmoid and GP models, we tested various biomarker hypotheses by examining the association between the overall response statistics from the models with genetic variants detected in the cell lines using a frequentist and a Bayesian approach (*Figure 1F–H*). For one biomarker hypothesis, as an example, we examined copy number alterations and point mutations in breast cancer cell lines in relation to the measured drug response of afatinib in those cells. The GP and sigmoid estimated $IC_{50}$ from cell lines treated with afatinib were significantly different in cases with and without *ERBB2* amplification (ANOVA q-value = 4.12e-9; *Figure 1I*). The GP models provided an added benefit of providing uncertainty estimates that were incorporated into a Bayesian hierarchical model to further verify the association between *ERBB2* amplification and afatinib sensitivity (posterior probability = 0.001; *Figure 1J*).

## Gaussian Processes provide estimates of dose-response uncertainty for single experiments

Both GP and sigmoid curve fitting produced comparable $IC_{50}$ and AUC estimates. Precursor sigmoid curve fitting methods based on Markov Chain Monte Carlo simulations enabled error estimates in $IC_{50}$ values (*Garnett et al., 2012*), however, this was neglected in the state-of-the-art sigmoid curve fitting (*Vis et al., 2016*) due to missing propagation to biomarker identification. Here, we introduce the added benefit of sampling from the GP posterior, which provides the models in-build uncertainty obtained for these $IC_{50}$ estimates. This is important for high-throughput drug screening experiments where there is often a high number of drugs and samples tested but very few replicate experiments. By applying the GP model to each experiment, we estimated the standard deviation for each $IC_{50}$ or AUC value based only on data points from that single experiment. These single sample standard deviations were compared to the standard deviations measured from here provided replicate experiments, that is the same drug tested multiple times on the same cell line and at the same concentration. We applied our GP estimation method to data from replicate experiments of 26 drugs on 10 cell lines, which contained 260 test conditions and 8 to 9 replicates for each condition. We wanted to see if an estimate of the uncertainty of the summary statistic, such as the standard deviation of the $IC_{50}$ posterior samples, would be correlated with the dispersion between replicates. Here, we refer to the variability between (mean) estimates for replicates as the observation uncertainty, and the variability in the estimate for a single replicate as the estimation uncertainty.

We compared observation and estimation uncertainty across replicate experiments of all 260 conditions (*Figure 2A*). When the estimation uncertainty is large, we will have less confidence in the estimated $IC_{50}$ in an experiment. Measurement errors for individual points in a dose-response curve will generally result in larger estimation uncertainty, whereas greater variation between biological replicates will result in larger observation uncertainty. We found two trends in the relationship between observation and estimation uncertainty. First, for experiments where the estimated $IC_{50}$ lies within the concentration range tested, the estimation uncertainty is positively correlated (Pearson correlation = 0.84, 95% CI [0.76, 0.89]) with the observation uncertainty. Second, for experiments where the estimated $IC_{50}$ lies beyond the maximum tested concentration, we observed a negative correlation (Pearson correlation = −0.39, 95% CI [−0.51,−0.25]). We note that the latter experiments

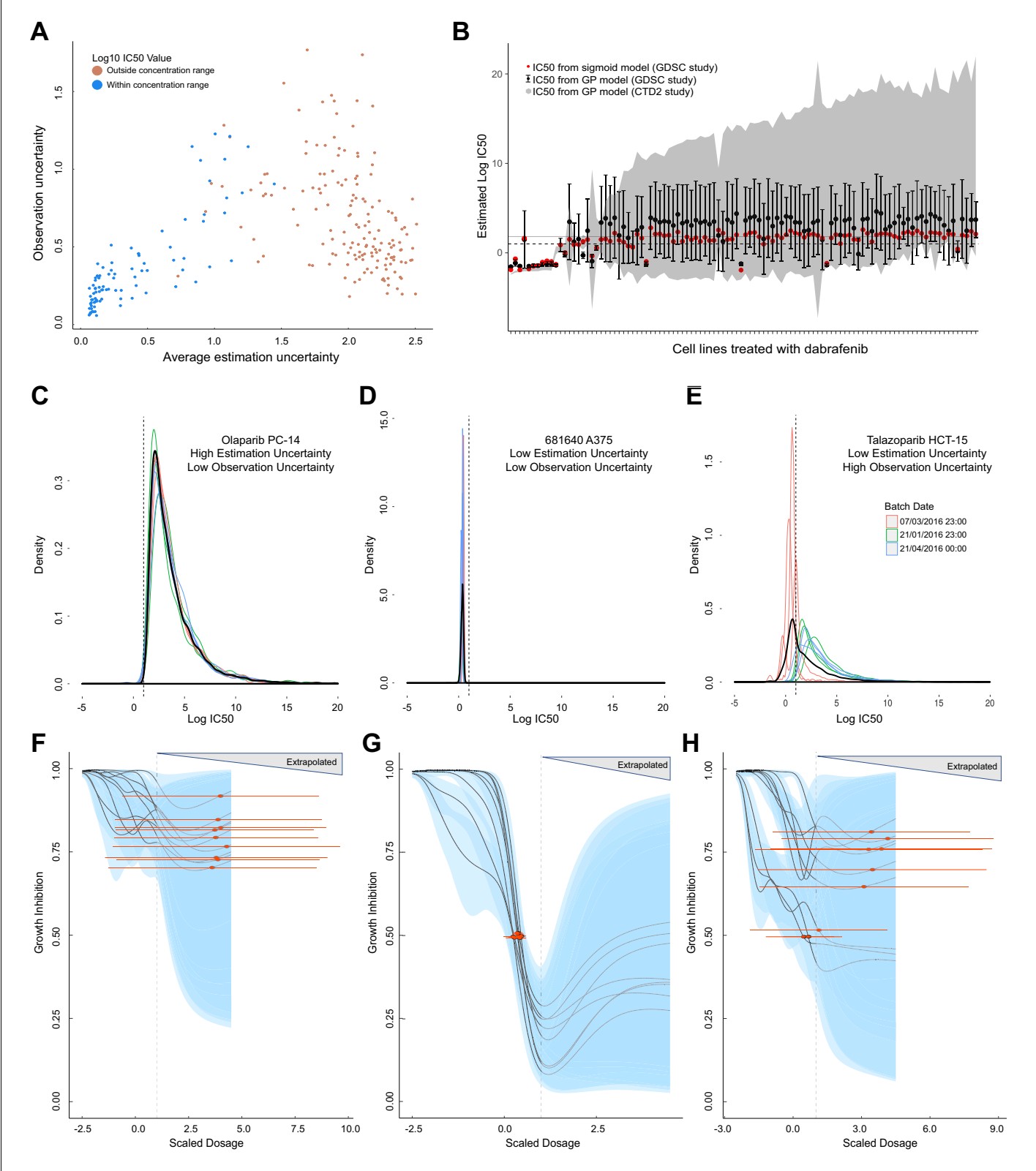

**Figure 2.** Comparison of GP estimates of uncertainty to replicate drug screening experiments. (**A**) Comparison between observational uncertainty (standard deviation over replicates of log10(IC$_{50}$) mean estimates) and estimation uncertainty (average over replicates of log10(IC$_{50}$) standard deviation) from each replication experiment. The colour of the points indicates whether the log10(IC$_{50}$) mean estimates were within or outside the maximum concentration range for each assay. (**B**) Mean IC$_{50}$ and the estimation uncertainty from the GPs for a BRAF inhibitor (dabrafenib) tested in each cell line

*Figure 2 continued on next page*

*Figure 2 continued*

in two independent studies (GDSC and CTD2). Estimation uncertainty (error bars and grey shading) were larger beyond the max concentration in both GDSC (dashed line) and CTD2 (grey line). The point estimates of the $IC_{50}$s from the GPs (black dots) were also comparable to the published $IC_{50}$s (red dots). (C-E) Three sets of replicate experiments, representing different amounts of estimation and observation uncertainty. Each density represents the distribution of $IC_{50}$ values from the Gaussian process samples from each replicate experiment. The colours represent different experimental batches. Narrow distributions demonstrate low estimation uncertainty and overlapping distributions demonstrate low observation uncertainty. The thick black line represents the density obtained by pooling samples from all replicates and the dashed line shows the maximal dosage tested. GP-curve fits corresponding to the three sets of replicate experiments showing $IC_{50}$ estimates with (F) high uncertainty, (G) low uncertainty, and (H) mix of uncertainties depending on whether estimates are made within or beyond the max concentration. The blue areas represent the 95% confidence interval in the curve fits and extrapolated GP curves (light grey lines) are displayed up to five times the maximum concentration, where the uncertainty will be extremely high.

The online version of this article includes the following figure supplement(s) for figure 2:

**Figure supplement 1.** Investigation of batch effects in the replicate data.

---

require extrapolation to estimate the $IC_{50}$ beyond the concentration range, which increases the estimation uncertainty, but does not generally affect the observational uncertainty. However, we observed that the estimation uncertainty from our GPs for dabrafenib (BRAF inhibitor) tested in two independent studies on the same cell lines were comparable both within and beyond the concentration range (*Figure 2B*).

Since the replicate experiments were conducted in batches over a period of several months, we verified that the observed trends held regardless of batches (*Figure 2—figure supplement 1*). Additionally, we examined the relationship between estimation uncertainty and observation uncertainty in a number of edge cases where $IC_{50}$ was estimated within and beyond the maximum concentration tested (*Figure 2C–E*). In the case of olaparib tested on PC-14, the uncertainty for the $IC_{50}$ within each replicate experiment was high, and this level of uncertainty was consistent across all replicates even beyond the max concentration (*Figure 2C and F*). In other replicate experiments, both estimation and observation uncertainty were low (*Figure 2D and G*), or varied depending on whether the batch reported mostly $IC_{50}$ values beyond the concentration range. Talazoparib tested in colorectal cancer line HCT-15 is a case where observation uncertainty was high, even though estimation uncertainty was low, and experiments in different batches showed different estimated $IC_{50}$s from very different dose-response curves (*Figure 2E and H*).

In order to examine the diversity of uncertainty estimates across experiments further, we described the relationship between AUC value of GP fits with their corresponding estimation uncertainty (*Figure 3*). We decided to use AUC here due to the greater uncertainty of estimating $IC_{50}$s beyond the maximum dose concentration. Since AUCs were computed within the tested concentration range, the estimation uncertainty for AUC was not substantially higher for cases where $IC_{50}$s were estimated within compared to beyond the maximum concentration (*Figure 3—figure supplement 1A*). The difference between the AUC estimates from the GP compared to the published GDSC sigmoid curve fits was greatest for experiments showing a partial response (AUC between 0.4 and 0.9), whilst at the same time these experiments also had the highest estimation uncertainty (*Figure 3A*). Our visual examination of the raw dose-response data from those experiments revealed evidence of poor quality readouts, for instance, where cell viability increases with increasing drug dose (*Figure 3—figure supplement 1B*). We were able to quantify the quality of these readouts by estimating the Spearman correlation coefficient based on the raw cell viability counts and the dose concentrations (*Figure 3B*). A negative Spearman correlation indicates that cell viability decreases as dosage increases (as expected) whilst a positive Spearman correlation indicates the opposite. The experiments with high estimation uncertainty from our GPs were also the experiments with high Spearman correlation pointing to poor quality.

Next, we investigated whether there were any attributes of experiments that would correspond to high estimation uncertainty and poor quality results. Labelling of experiments based on cell culture conditions, dose and cancer type revealed no obvious associations with estimation uncertainty (*Figure 3—figure supplement 2A–E*). However, there was a large spread in the uncertainty estimates for AUC when we grouped the experiments into target pathways based on the primary targets of the tested drugs (*Figure 3C*; *Figure 3—figure supplement 2F*). Whilst most drugs had similar average AUC point estimates between 0.6 and 0.8, suggesting they all had a spread of

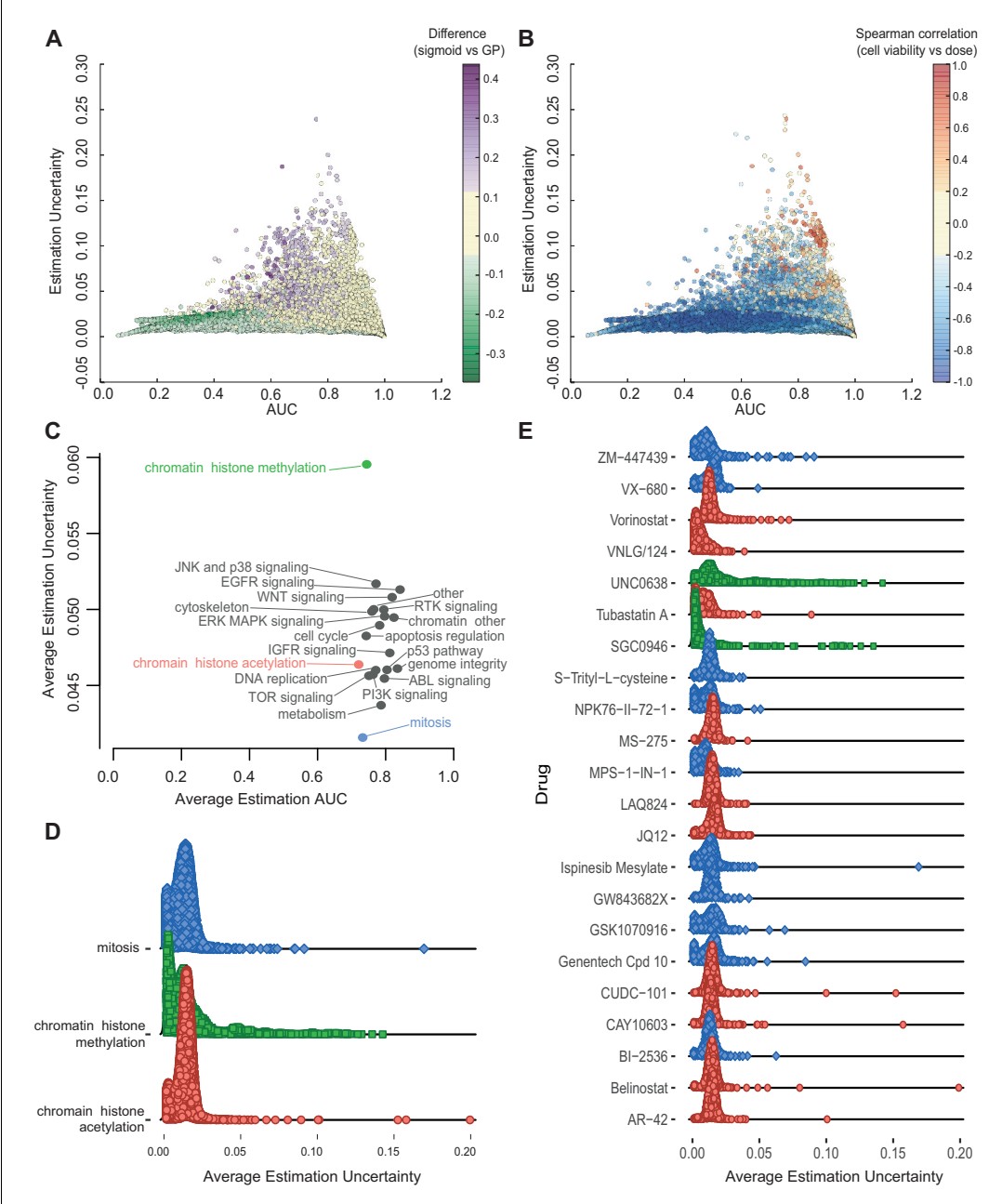

**Figure 3.** Relationship between AUC and uncertainties estimated from GPs across all experiments. (A) Coloured by difference between the AUC estimated by sigmoid vs GP fits. (B) Coloured by Spearman correlation between cell viability and dose concentration in the raw data. Poorer experiments (orange-red) result in greater uncertainty and positively correlated with cell viability increasing with higher dose. (C) Average uncertainty and AUC for experiments with uncertain fits (estimation uncertainty >0.03) with drugs grouped by their target pathway. (D) Distribution of estimation uncertainty for all drugs targeting chromatin histone methylation, chromatin histone acetylation, and mitosis and (E) for individual drugs. The online version of this article includes the following figure supplement(s) for figure 3:

**Figure supplement 1.** High estimation uncertainty independent of concentration range.
**Figure supplement 2.** Relationship between uncertainties and other experimental factors.

experiments showing resistance and sensitivity, the average estimation uncertainties varied across target pathways. Interestingly, similar target pathways (e.g. chromatin histone methylation and chromatin histone acetylation) had very different levels of estimation uncertainty. Within each of these target pathways, we also see different distributions of estimation uncertainties (*Figure 3D*). Most

target pathways have a bi-modal distribution representing compounds that have low uncertainty in the cases of clear sensitivity or resistance, and high uncertainty in the cases of partial responders (*Figure 3E*). Both chromatin histone methylation drugs in particular had a much longer right tail towards higher estimation uncertainties that are associated with poor experimental readouts, or possibly off-targets.

## Curve fits using Gaussian Processes can help identify clinically relevant biomarkers

The $IC_{50}$ values are highly conconcordant for sigmoid and GP-curve fittings, showing an average weighted Pearson correlation of 0.88 (95% CI [0.85; 0.91]) across individual drugs, and cancer types (*Figure 4A*). Strong agreement is found when true responding cell lines were observed in the screen (*Figure 4B*). For example, if >10% of cell lines responded within the concentration range, that is $IC_{50}$ <maximum tested concentration, then a weighted Pearson correlation >0.75 was consistently achieved for all drugs. We found positive correlations for all drugs, even when comparing exclusively non-responding cell lines, where all the $IC_{50}$ values are extrapolated beyond the maximum dosage range. Drug-response values are concordantly fitted with both methods for sensitive cell lines (*Figure 4C*, mean log10($IC_{50}$) in µM of 0.02 95% CI [−0.05; 0.09]), whilst extrapolated non-responders tend to lead to more conservative and higher $IC_{50}$ values fitted with GP (*Figure 4C*, mean log10 ($IC_{50}$) in µM of 1.10, 95% CI [1.03; 1.18]). Whilst the average fits from the sigmoid and GP models identify known clinical biomarkers, there are clearly differences for individual cell lines, especially when the $IC_{50}$ value has been extrapolated beyond the dosage range, that may help identify new biomarkers. Alternatively, AUC values can be used to compare both curve fitting methods (*Figure 4—figure supplement 1*). Whilst known clinical biomarkers are recovered with AUC as a drug-response metric, $IC_{50}$ measures were used in the subsequent analysis as they retain direct relationship with the drug concentration and are more interpretable.

To highlight the overall agreement of both curve fitting methods, we systematically tested 26 clinically established biomarkers of drug response (*Figure 4D*, *Figure 4—figure supplement 2A–C*, *Supplementary file 1*) using previously established association tests (*Iorio et al., 2016*), 24 of which were significantly reproduced regardless of sigmoid or GP-curve fitting (10% FDR). For example, both curve fittings captured the association of BRAF inhibitors (PLX4720, progenitor of vemurafenib; and dabrafenib) with *BRAF* mutations in melanoma (*Figure 4—figure supplement 3A–C*; *Chapman et al., 2011*). Dabrafenib is a potent BRAF inhibitor and in addition we detected *BRAF* mutations as a sensitivity marker in thyroid carcinoma (*Figure 4D*, *Figure 4—figure supplement 3D*). Another example are the EGFR inhibitors, afatinib and gefitinib, that are concordantly correlated with drug sensitivity in *EGFR* mutant cell lines in lung adenocarcinoma (*Figure 4—figure supplement 3E–G*; *Tamura and Fukuoka, 2005*; *Yang et al., 2012*). *ERBB2*(*HER2*) amplification in breast cancer was also recapitulated as a biomarker of sensitivity to the dual EGFR/ERBB2 inhibitor lapatinib (*Figure 4—figure supplement 3H*; *Konecny et al., 2006*). Among the 26 clinical biomarkers, we consistently found drug resistance of *TP53* mutants to MDM2 inhibition with nutlin-3a in five different cancer types (*Figure 4E*, *Figure 4—figure supplement 3I–L*). Overall, the majority of expected clinical and preclinical biomarkers are reproduced, regardless of the drug-response curve fitting method.

We concordantly and significantly identified six novel (not yet clinically established) drug sensitivity biomarkers (0.1% FDR) regardless of the applied drug-response curve fitting method. Investigating two different curve fitting algorithms, and retrieving the same biomarkers can be considered as a test of robustness, which in our case concordantly highlighted non-gold standard associations for prioritising experimental validation. For example, daporinad (also known as FK866 and APO866) is a small-molecule inhibitor of nicotinamide phosphoribosyltransferase leading to inhibition of NAD+ biosynthesis. It has been clinically tested in melanoma (ClinicalTrials.gov Identifier: NCT00432107), Refractory B-CLL (NCT00435084) and Cutaneous T-cell Lymphoma (NCT00431912), whilst showing anti-proliferative effect in glioblastoma cell lines (*Zhang et al., 2012*). Therapeutic potential when combining with other drugs used to treat gliomas (*Lucena-Cacace et al., 2019*; *Lucena-Cacace et al., 2017*) has been suggested, whilst we additionally and concordantly identify *EGFR* amplification as a biomarker (*Figure 4F*).

Another novel and concordant identified biomarker is doramapimod response (also known as BIRB-796) in *ARID2* mutant melanoma cell lines (*Figure 4G*). Doramapimod is a small-molecule p38

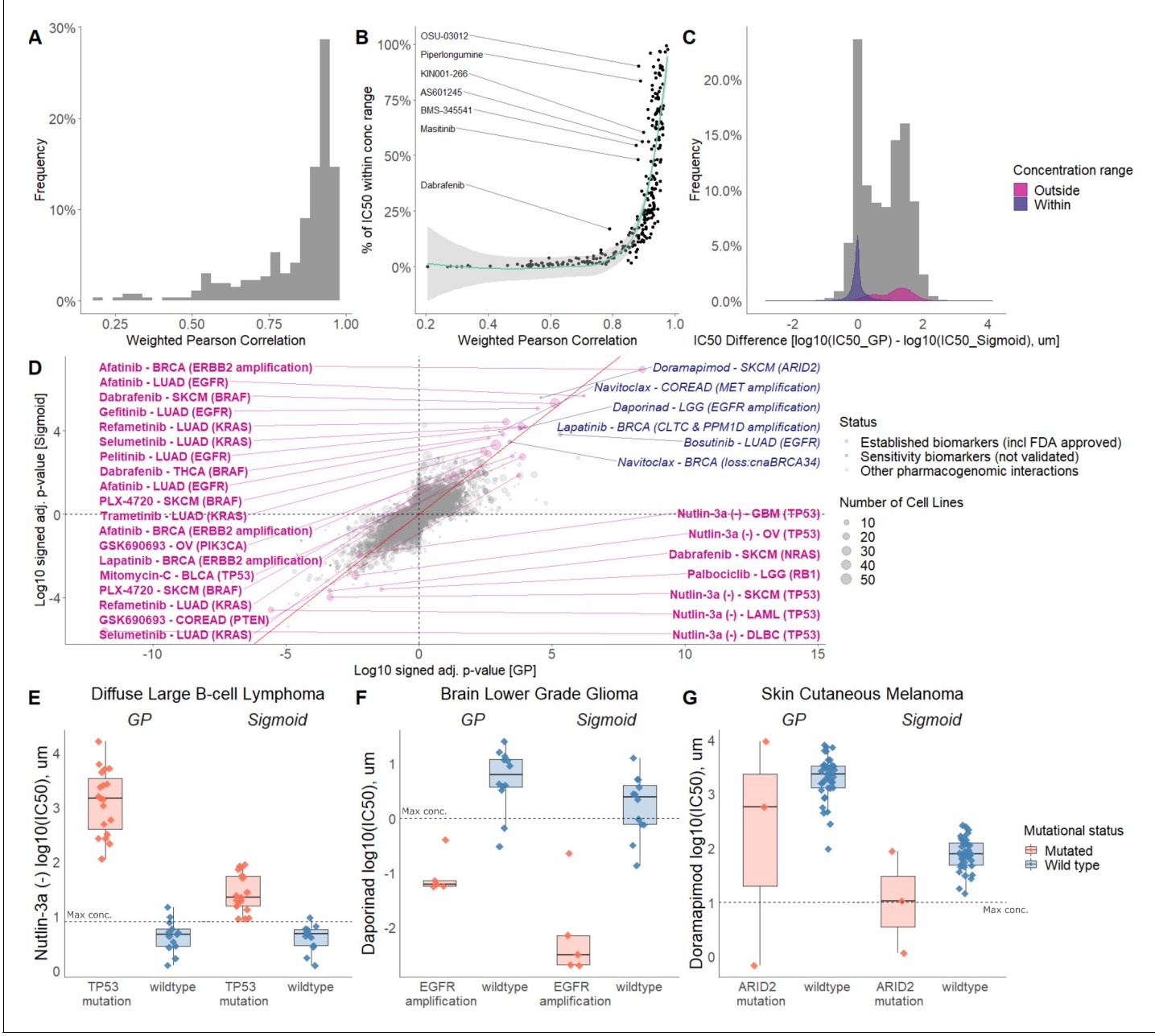

**Figure 4.** Comparison of sigmoid and GP-curve fitting. (A) Weighted Pearson correlation of each drug within cancer types. (B) Comparing the concordance of sigmoid and GP-curve fitting when stratifying for percentage of cell lines with $IC_{50}$ value lower than maximum concentration. (C) $IC_{50}$ value difference between GP and sigmoid curves. Grey histogram represents frequency distribution of the $IC_{50}$ value difference between GP and sigmoid curves without stratification by within/outside the concentration range. (D) Drug-response biomarker comparison based on both curve fittings (sigmoid vs GP). The Benjamini-Hochberg adjusted p-values are in log10 scale and signed based on the direction of the effect size (Cohen's d). Additional biomarker examples for (E) diffuse large B-cell lymphoma (DLBCL) treated with nutlin-3a (MDM2 inhibitor) and stratified by *TP53* mutants; (F) Low grade glioma (LGG) treated with daporinad (NAMPT inhibitor) and stratified by *EGFR* amplification; (G) Skin cutaneous melanoma (SKCM) treated with doramapimod (p38 and JNK2 inhibitor) and stratified with *ARID2* mutations.

The online version of this article includes the following figure supplement(s) for figure 4:

**Figure supplement 1.** Comparison of GP and Sigmoid curve fitting using AUCs.

**Figure supplement 2.** Comparison of sigmoid and GP-curve fitting using $IC_{50}$s.

**Figure supplement 3.** Drug-response biomarker comparison based on both curve fittings.

MAPK inhibitor and has been reported in different cancer types (in combination with other drugs) including cervical cancer, paracrine tumours and myeloma (*Jin et al., 2016*; *Yasui et al., 2007*). ARID2 is part of chromatin remodelling complex and is involved in DNA repair in hepatocellular carcinoma cells (*Oba et al., 2017*) and enriched in melanomas (*Ding et al., 2014*; *Hodis et al., 2012*). In conclusion, different curve fitting approaches lead to concordantly and novel identified biomarkers, thereby increasing the robustness in those findings, and consequently enabling to prioritise hypotheses.

## Improved biomarker detection by taking into account uncertainty in a Bayesian framework

Since both Bayesian and frequentist methods can be used to prioritise biomarkers for further testing, we compared association statistics (posterior probabilities and q-values) from both statistical methods. We observed a number of cases where the Bayesian and ANOVA tests disagree (*Figure 5A*; *Supplementary file 2*). For instance, *BRAF* mutations in colorectal cancer were detected as a sensitivity biomarker for dabrafenib by the Bayesian test, but less significant by the ANOVA test. This association had been repeatedly reported in in vitro models (*Iorio et al., 2016*; *Rees et al., 2016*) and also found in melanoma cases (*Chapman et al., 2011*), whilst not in colorectal cancer patients due to feedback activation of ERK-signalling mediated via *EGFR* (*Corcoran et al., 2018*; *Prahallad et al., 2012*). We note in *Figure 5B* that the Bayesian test takes advantage of the additional information that sensitive mutant cell lines have low estimation uncertainty, whilst the small number of resistant mutant cell lines have high estimation uncertainty, causing them to have less influence on the biomarker detection. On the other hand, the ANOVA model detected the *KRAS* copy number alteration as a resistance biomarker for lenalidomide (immunomodulatory drug) partial sensitivity in skin cutaneous melanoma (SKCM), whilst not detected by our Bayesian approach. Whilst on the linear $IC_{50}$ scale there is some difference between the small number of mutant cell lines and wildtypes, the Bayesian model considered that the estimated responses of the mutant cell lines had high uncertainty (*Figure 5C*). Additionally, a comparison of the uncertainty estimates for the GP and the Sigmoid curve fitting methods revealed that both display concordant results (*Figure 5B and C*; *Figure 5—figure supplement 1*); However, the Sigmoid curve fitting method (Materials and methods; *Vis et al., 2016*) underestimates variance of non-responding cell lines rendering the GP approach superior. The dosages within *Figure 5B and C* were rescaled to prevent the need for adapting the length-scale hyperparameter to the maximum dosage. $IC_{50}$ values were back-transformed to the log10 drug dosage scale to make comparisons with (*Iorio et al., 2016*) (see Materials and methods). Whilst discrepancies between Bayesian and ANOVA tests have to be taken with caution, they may highlight novel biological insights which would be missed when applying only a single model.

## Discussion

The GP approach developed in this research has several advantages compared to the traditional approach of fitting sigmoidal drug-response curves. Firstly, these flexible, non-parametric models can be used to fit a wider variety of dose-response curves than the parametric sigmoidal models, for example curves of unexpected shapes may reflect biological signals of off-target effects. Secondly, the GP models provide straightforward uncertainty quantification of any summary statistic that can be calculated on a dose-response curve, a fact that we take advantage of in developing our hierarchical Bayesian model for biomarker testing. Thirdly, the GP model can deal with outlying measurements better than a sigmoidal model, due to formulating it as a mixture model with one component representing the latent GP process of the drug response, and the second component accounting for outliers.

In contrast to other GP-based models in *Tansey et al., 2018*, our approach is highly interpretable, as we do not integrate the biomarkers into the model estimation in a non-linear fashion, but instead proceed in a two-step approach that first fits our Gaussian process model to the dose-response curves, and then uses the derived summary statistics and uncertainty measures to perform biomarker detection. Thus, we can take advantage of the flexibility of the Gaussian process without the complexity of fitting a non-linear neural network to enable prediction from molecular measurements.

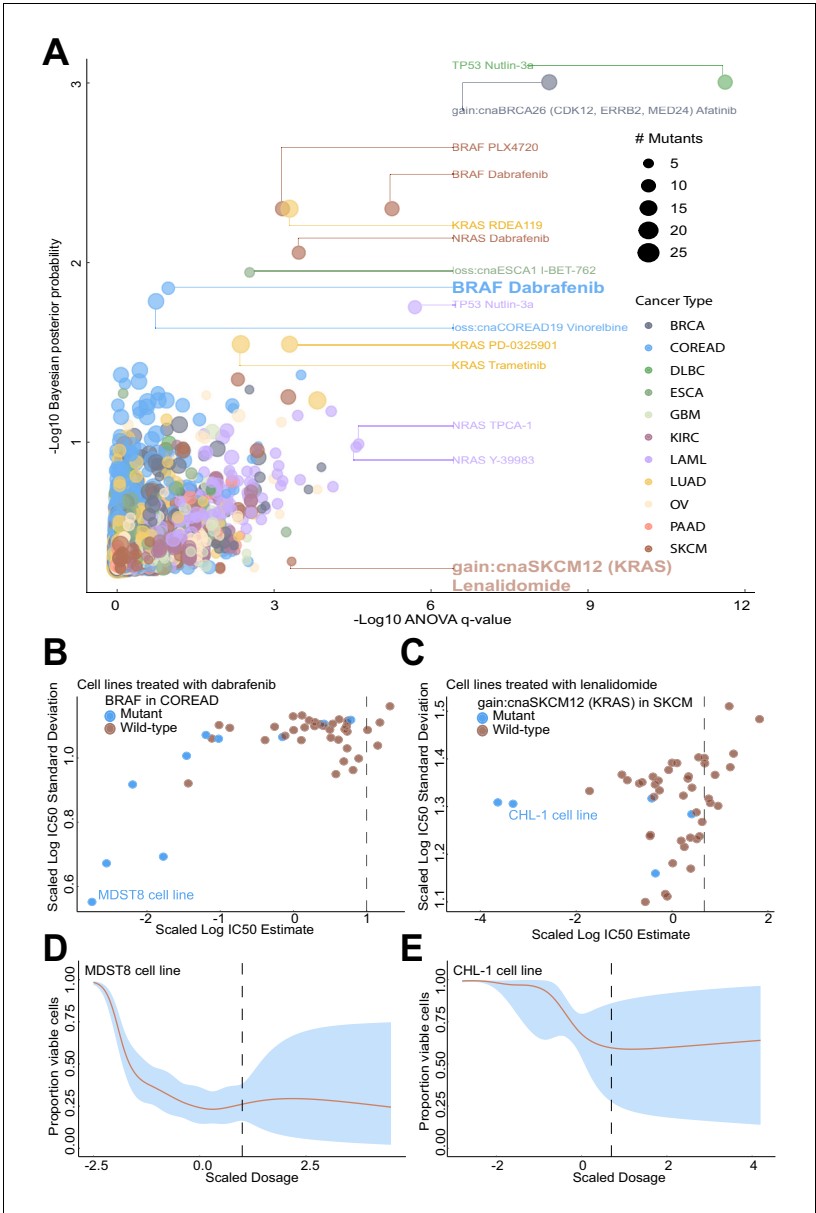

**Figure 5.** Comparison of Bayesian testing and ANOVA using the GP IC$_{50}$ estimates. (**A**) Scatterplot of biomarker associations with IC$_{50}$ drug response. The y-axis shows the negative log10 transformed posterior probability of a sign change in the effect under the Bayesian testing model, whilst the x-axis shows the negative log10 of the q-value from ANOVA testing. The size of the circles is proportional to the number of mutants or copy number variations in the given type of cancer cell line. (**B**) GP estimates for the mean and standard deviation of the log (IC$_{50}$) from colorectal cell lines tested with BRAF inhibitor dabrafenib, which showed significant association with *BRAF* mutation in the Bayesian test. (**C**) Estimated IC$_{50}$ and its uncertainty for skin cutaneous melanoma cell lines tested with the immunomodulatory drug lenalidomide, which showed significant association with *KRAS* copy number alteration in the ANOVA test. Black vertical lines show the location of the maximum experimental drug dosage. Dose-response curve of the (**D**) MDST8 colorectal cancer cell lines with *BRAF* mutation treated with dabrafenib. The black dotted line represents the maximum concentration of the drug used to treat the cell lines. The blue area represents the 95% confidence intervals in the dose-response fits. (**E**) Similar to (**D**) but for CHL-1 skin cutaneous melanoma cell lines with *KRAS* copy number alteration treated with lenalidomide.

The online version of this article includes the following figure supplement(s) for figure 5:

**Figure supplement 1.** Sigmoid curve fitting uncertainty.

**Figure supplement 2.** An overview of the cell viability values.

The increased flexibility of the GP model comes at a price. Most notably, because we do not impose a specific functional form, there are few constraints on the behaviour of the curve outside the range of observed dosages. This leads to the counter-intuitive behaviour that the posterior mean estimate of drug response can go up when extrapolating beyond the maximum dosage. Note, however, that this goes along with a commensurate increase in the posterior variance (*Figure 5D,E*). In other words, the model is highlighting that extrapolation beyond the observed dosage range is highly uncertain, and the posterior mean estimate should not be relied on. It would be possible to constrain this behaviour by introducing artificial data points at a high concentration, or less crudely by imposing monotonicity constraints via virtual derivative observations (*Riihimäki and Vehtari, 2010*). However, these methods would limit the flexibility of our method and lead us to underestimate the uncertainty of the posterior mean. An alternative approach is to constrain the Gaussian process using generalised analytic slice sampling (*Tansey et al., 2019*), which integrates the constraints into the sampling process. Whilst theoretically appealing, this approach is not compatible with the variational inference method that we have chosen for our work, and would lead to an unacceptable increase in computational burden for fitting the dose-response curves.

We have systematically compared the application of GP to sigmoid models across a pan-cancer drug screen. We demonstrated that our GP estimates of the $IC_{50}$ values and their subsequently predicted biomarkers using ANOVA are reliable when compared to estimates from the sigmoid models. In addition, the GP models provide useful information about the uncertainty associated with the drug-response quantification. However, there is a crucial difference between estimation uncertainty on a single experiment and observational uncertainty across multiple replicates of the same experiment, which incorporates measurement error, technical and biological variation. We are interested in the former to assess the quality of the fit, and therefore the reliability of the estimated $IC_{50}$. We hypothesized that estimation uncertainty characterises observational uncertainty within the dose concentration range tested. However, extrapolating beyond the concentration range would be challenging due to the uncertainty in the behaviour of the dose-response curve in unobserved concentrations. Imposing monotonicity may not be the best path in this case, but we avoid making this assumption. Instead, our method defines a very large confidence interval for drug-response statistics extrapolated beyond the maximum dose tested and we would additionally need to take the observation uncertainty between replicate experiments into account. We have verified this by applying our estimation method to a replication data set of 26 drugs tested on 10 different cell lines, with 8 to 9 replicates for each drug-cell line experiment. We conclude that whilst estimation uncertainty is a useful indicator for within-concentration $IC_{50}$ values, it cannot be used as a proxy for observation uncertainty when the $IC_{50}$ is extrapolated beyond the tested concentration range. Indeed, overall drug responses and biomarkers from independent drug screens were consistent when comparing similar dose ranges (*Haverty et al., 2016*). Any difference between replicate experiments may be due to batch effects or other unobserved factors that are not necessarily reflected in the estimation error. Whilst previous studies have attempted to capture uncertainty by measuring the spread of the residuals from the fitted curves, such as root mean square error, they were not able to capture these false positive biomarkers by setting strict cutoffs (*Cokelaer et al., 2018*).

Whilst Bayesian posterior probabilities and ANOVA q-values are different statistical quantities for measuring biomarker associations that should not be compared in absolute terms, we compared these quantities in relative terms to prioritise biomarkers of response for further testing. Our Bayesian biomarker model extends the classical ANOVA testing, since it is able to leverage the estimation uncertainty of the $IC_{50}$ values. We showed that taking estimation uncertainty into account in the Bayesian model can lead to both inclusion and exclusion of putative biomarkers. For example, the Bayesian model highlighted the association between *BRAF* mutation in colorectal cancer and BRAF inhibitor response. Targeting BRAF signalling has recently been confirmed as a viable option for metastatic colorectal cancer cases with *BRAF* mutations (*Kopetz et al., 2019*). In contrast, the Bayesian model excluded a suggestion from ANOVA of association between *KRAS* mutation with lenalidomide response in melanoma. Lenalidomide has thus far had no clinical success in *KRAS* mutant cases nor melanoma (*Gandhi et al., 2013*; *Glaspy et al., 2009*).

Although we systematically tested for drug-biomarker associations, we did observe common behaviour for certain cell types or classes of drugs. The high uncertainty in the response estimates of chromatin histone methylation targeting compounds for instance may be due to the large number of factors contributing to epigenetic regulation of cells (*Luo, 2015*). It would be straightforward to

extend the GP model to allow for sharing information across drugs or cell lines of similar class, by using either shared hyperparameters or a hyperprior on the hyperparameters. We have not implemented this approach in our work here as our aim was to show the advantage of fitting individual drug-response using GPs, and extending the method to fitting multiple curves jointly would increase the memory and computational requirements significantly. It is our hope to continue expanding the suite to multiple dimensions of dose-response and biomarker prediction needed for drug combinations, which is predominantly based on synergy modelling with either Loewe Additivity or Bliss Independence (*Di Veroli et al., 2016*; *Vlot et al., 2019*). In cases where multiple statistical models converge to concordant biomarkers, this increases the reproducibility of the evidence, potential for clinical translatability and ultimately enables precision medicine.

The increasing utilisation of high-throughput drug screening for identifying effective new treatments will necessitate the use of more powerful statistical and machine learning methods (*Toh et al., 2019*). We have introduced an approach for quantifying the uncertainties of dose-response using Gaussian Processes and further described how these uncertainties can be integrated into statistical testing of biomarkers. For cancer treatments, our approach can help estimate the uncertainty of dose-responses reported in the numerous drug screening studies by academic (*Ghandi et al., 2019*; *Holbeck et al., 2017*; *Iorio et al., 2016*) and pharmaceutical laboratories (*Menden et al., 2019*; *O'Neil et al., 2016*). This can provide more robust metrics for comparing drug responses to identify the most potent ones and highlight sensitivity biomarkers that are more likely to succeed clinically because they are associated with low uncertainty. The approach is also generalisable beyond cancer to any disease and any dose-response measures. We hope that by considering response uncertainty and providing a probabilistic view of drug biomarkers, the risks associated with drug development can be better balanced and smarter decisions can be made.

# Materials and methods

## Key resources table

| Reagent type (species) or resource | Designation | Source or reference | Identifiers | Additional information |
|---|---|---|---|---|
| Cell line (*Home sapines*) | 1074 cancer cell lines | (*Iorio et al., 2016*) PMID:27397505 | GDSC cell line drug response:GDSC1 (v17); GDSC cell line genomics: GDSCtools_mobems | Further information about the cancer cell lines from the GDSC can be found here: https://www.cancerrxgene.org/downloads/bulk_download |
| Software, algorithm | Source code for curve fitting and Bayesian biomarker detection | This paper | | All source code can be found via GitHub here: https://github.com/FrankD/GPDrugModels |
| Software, algorithm | GPFlow | GPFlow (https://www.gpflow.org) | | Version 1.5.1 |
| Software, algorithm | TensorFlow | TensorFlow (https://www.tensorflow.org/) | | Version 1.14.0 |

## Drug screening

We analysed 1074 cancer cell lines tested with 265 compounds from a high-throughput screen resulting in 225,384 experiments that were previously published (*Iorio et al., 2016*). Cell line data was retrieved and is publically available via the GDSC website (Key Resources Table). All cell lines were authenticated. Details for each cell line can be found at: https://www.cancerrxgene.org/help.

Compounds were tested with 5 to 9 titration points, whilst either diluted with 4- or 2-fold, respectively. Cells were seeded on day zero, left in the microtiter plate for 24 hr to retain linear growth, and consecutively treated for 3 days. After those 3 days of treatment, cellTiterGlo staining is used to quantify ATP levels within each well. In parallel, untreated cells and blank wells were also measured to estimate and normalise cell viability.

Compounds within the replicate study were screened across a seven point dose-response curve with a half-log dilution and 1000 fold range. The duration of drug treatment was 72 hr and cell viability was measured using CellTiter-Glo (Promega). Each cell line and compound pair was screened in

technical triplicate, three assay plates generated simultaneously, and across three biological replicates with 46 and 44 days between the first to second and second to third replicates respectively. Cell viability measurements for these experiments can be found in *Supplementary file 3*.

## Preprocessing

Prior to analysis, we scaled the raw observed fluorescent intensities for each drug/cell line combination using the observations from the blank and negative control wells as follows. Let $R = \{r_1, r_2..., r_n\}$ be the observed intensities for *n* dosages. Let *B* be the mean of the intensities for the blank wells on the same plate as the experiment, and C be the mean of the intensities of the negative control wells (no drug added). Then the relative cell viability *V* can be calculated as:

$$V = \frac{R - B}{C - B}$$

Relative cell viability values below 0 (n = 2646, *Figure 5—figure supplement 2*) were set to 0.

For the purpose of fitting the Gaussian process models, we additionally rescale the dosages to avoid having to adapt the length-scale hyperparameter to the maximum dosage. We rescale the $\log_2$-transformed dosages $d = \{d_1, d_2..., d_n\}$ as follows:

$$d' = \frac{d + 1}{max(d) + 1}$$

Note that $IC_{50}$ values have been back-transformed to the log10 drug dosage scale for comparability with those reported in *Iorio et al., 2016*.

## Sigmoid drug-response model

The GDSC estimates in *Iorio et al., 2016* were obtained using a sigmoid fit to the drug-response curve, using the same pre-processing of the fluorescent intensities as described above. The particular sigmoid model used is the one described in *Vis et al., 2016*. In brief, if we have shape parameter $s_i$ and position parameter $p_{ij}$ for cell line *i* and drug *j* , then cell viability can be represented as a function of dosage *d*:

$$f(d, s_i, p_{ij}) = \frac{1}{1 + exp(\frac{d - p_{ij}}{s_i})}$$

Note that this allows for cell line/drug specific position parameters, but shape parameters that only vary by cell line and are shared across drugs. The position parameter $p_{ij}$ corresponds to the estimated $IC_{50}$ for cell line *i* and drug *j*. For full details, see *Vis et al., 2016*.

To estimate the uncertainty of the Sigmoid curve fitting, a random bootstrap sampling of 80% of all treated cell lines available for each drug over 100 iterations was performed. The Sigmoid curve fitting model from GDSC (Vis et al. 2013) estimates one scale parameter per drug across all treated cell lines, thus the sampling creates variance in the response data. The standard deviation of the log $(IC_{50})$ estimates was computed to assess the model's variance.

## Gaussian process drug-response model

For simplicity, we drop the subscripts *ij* and present the combination. We model the drug response **y** via a two-component Beta mixture such that:

$$P(\mathbf{y}|\mathbf{f}, s_1, \mu_2, s_2, \pi) = \pi \, Beta^{\mu}(\mathbf{y}|\Phi^{-1}(\mathbf{f}), s_1 + (1 + \pi) Beta^{\mu}(\mathbf{y}|\mu_2, s_2))$$

where $Beta^{\mu}$ is the reparameterization of the Beta distribution in terms of the mean μ and a scale parameter *s*, and $\Phi^{-1}$ is the probit function (the inverse of the standard normal cumulative distribution function). Component one represents the drug response, which is driven by a latent Gaussian process **f**, whilst component two represents outliers that deviate from the overall dose- response trend. We set the scale parameters $s_1 = 50$ and $s_2 = 11$ and specify $\mu_2 = 0.9$ to reflect our belief that outliers will mostly be erroneous measurements of resistance. We set $\pi = 0.999$ as we believe that outliers are rare.

We place a standard Gaussian process prior on **f**, such that:

$$P(\mathbf{f}|d, \Psi) = MVN(\mathbf{f}|\mathbf{m}, C_\Psi(d, d'))$$

where $\mathbf{m}$ is the mean drug response, and $C_\Psi(d, d')$ is a covariance function with hyperparameters $\Psi$; in practice we choose a combined linear-Matern3/2 as a flexible option, which avoids the excessive smoothness of restrictions of the commonly used RBF kernel. Stein (1999) argues that this is a more realistic representation for physical processes (*Stein, 2012*). Information sharing across drugs and cell lines can be achieved via shared hyperpriors in a hierarchical model. For the application in this paper joint inference with shared hyperpriors would be computationally difficult, and we choose to instead empirically set the variance and length-scale parameters for the Matern to 0.2 and 0.3, respectively, and the variance parameter for the linear kernel to 0.1.

Inference is performed using variational learning (*Hensman et al., 2013*), via the GPFlow software (*Matthews AG de and van der Wilk, 2017*). We choose variational learning over alternatives such as Markov chain Monte Carlo due to its speed, which allows us to process large drug-response panels in a realistic time frame. Hyperparameters for the GP model were determined by manual tuning; however, for other datasets, we could also envision a Bayesian model selection procedure which places the variational inference in a variational-within-MCMC scheme where the MCMC moves update the hyperparameters. If fixed hyperparameters are desired, one could use the maximum a posteriori values. To avoid massive computational complexity, the MCMC scheme could be run on a representative subsample of cell lines.

## Calculation of summary statistics

Summary statistics of drug response can be calculated straightforwardly by sampling from the posterior of the Gaussian process (*Supplementary file 4*). Generally, let $g(\mathbf{d}, \mathbf{y})$ be a function that calculates a summary statistic $\tau$ from a dose-response curve with dosages $\mathbf{d}$ and responses $\mathbf{y}$, then we can obtain a posterior estimate of the mean of the summary statistic by first sampling $N$ dose-response curves from the posterior of the GP model, and then calculating the average:

$$\bar{\tau} = \frac{1}{N}\sum_l^N g(\mathbf{d}_l, \mathbf{y}_l)$$

A similar procedure can be used to calculate the posterior estimate of the standard deviation.

Although we can extract other response statistics from our curve fits, the most common are the $IC_{50}$ and the area under the drug-response curve (AUC). On the $\log_2$ dosage scale the dosages are equally spaced, and hence AUC can be straightforwardly estimated by the mean function:

$$g_{AUC}(\mathbf{d}, \mathbf{y}) = \frac{1}{n}\sum_m^n y_m$$

where $m$ indexes over the $n$ dosages. For the $IC_{50}$, estimation for a single curve is complicated by the fact that the curve may not cross the 50% viability threshold within the observed dosage range (non-crossing sample). We therefore extrapolate the GP samples to 10 times the maximum ($\log_2$) experimental dosage and specify $g_{IC50}(\mathbf{d}, \mathbf{y})$ as:

$$g_{IC50}(\mathbf{d}, \mathbf{y}) = d_m \text{ such that } y_m = 0.5 \text{ if } \exists y_m \leq 0.5$$

Note that this ignores samples where for all dosages, $y_m \leq 0.5$; one could devise a multivariate sufficient statistic that takes this information into account, but we have found that in general there is a reasonable amount of correlation between $g_{IC50}(\mathbf{d}, \mathbf{y})$ and the number of non-crossing samples for a given cell line/drug combination.

## Comparison of GP and sigmoid $IC_{50}$ values

Concordance between $IC_{50}$ values based on sigmoid and GP-curve fitting is quantified with Pearson correlation for each drug. To account for tissue specificity and the varying number of cell lines assessed per tissue type, we employed the average weighted Pearson correlation ($pw$) of the sigmoid-curve versus GP-curve fitted $IC_{50}$ values for the individual cancer types ($i$).

The weight for a given cancer type $i$ was denoted as $\sqrt{n_i - 1}$, where $n_i$ is the total number of cell lines treated with the drug within this tissue type. The following metric was applied,

$$pw = tanh\left(\frac{\sum_{i=1}^{N} \sqrt{n_i - 1}\, arctanh(p_i)}{\sum_{i=1}^{N} \sqrt{n_i - 1}}\right)$$

where $p_i$ is unweighted Pearson correlation within a cancer type ($i$) and a total number of tested cancer types is $N = 30$. For a given drug and tissue type combination, at least 10 cell lines need to be treated ($n_i \geq 10$).

Differences in $IC_{50}$ values for each drug-response value $j$ were consistently defined as

$$df_i = IC50_{j,GP} - IC50_{j,sigmoid}$$

with a total number of tested cell line and drug combinations equalling to $N_j = 171,937$.

## Bayesian biomarker testing

Standard statistical approaches for testing the influence of biomarkers on drug response mostly rely on analysis of variance (ANOVA) testing. An ANOVA can be understood as a linear model of the dependent variable $i$ (in this case, a summary measure of drug response such as $IC_{50}$):

$$g_i = \alpha + \beta z_i + \gamma x_i + \epsilon_i$$

where $x_i$ is an indicator variable denoting the group membership of data point $i$. In our application, the data points are cell lines, $z_i$ indicates group membership, for example the mutation status of a given SNP, and $x_i$ indicates any other covariates that we wish to correct for, such as tissue type. The parameter $\alpha$ captures the global mean of the drug response, whilst $\beta$ captures the effect of mutation status on the drug response, $\gamma$ is the effect of covariates, and $\epsilon_i$ is independent Gaussian noise.

This model, whilst useful, fails to account for the fact that our Gaussian process model provides estimates $\sigma_i$ of the uncertainty (or standard error) associated with the mean $IC_{50}$ estimates $g_i$. In order to make use of these uncertainty estimates, we take an idea from Bayesian meta-analysis, and integrate them via a hierarchical model:

$$g_i \sim \mathcal{N}(\mu_i, \sigma_i^2)$$
$$\mu_i \sim \mathcal{N}(\alpha + \beta z_i + \gamma x_i, \sigma^{*2})$$

where $\mu_i$ is the mean drug-response estimate for cell line $i$, and $\sigma^{*2}$ is the variance across cell lines (the variance of $\epsilon_i$ in the ANOVA example). Note that this model can be reduced to:

$$g_i \sim \mathcal{N}(\alpha + \beta z_i + \gamma x_i, \sigma_i^2 + \sigma^{*2})$$

We further specify a Gaussian prior $\beta \sim \mathcal{N}(0, 0.1)$ on the effect size parameter to discourage false positives and reflect our prior belief that most mutations are not associated with drug response. We also place an exponential prior $\sigma^{*2} \sim Exp(10)$ to regularize the variance parameter. Finally, $\alpha \sim \mathcal{N}(0, \tau^2)$ is a Gaussian prior on the global mean with standard error $\tau \sim Gamma(1, 1)$. Early exploratory results showed that using the estimates of $\sigma_i$ directly placed too much weight on experiments with very low estimation uncertainty, leading to unrealistic posterior estimates of the effect size $\beta$. To attenuate this, we used a transformed estimate $\sigma_i^c$, where the effect of parameter $c$ was explored over the range [0,1], and empirically set to 0.25 for the results reported in this paper. The main tuneable hyperparameter is the scaling parameter c, as the model is robust to changes to the parameters for the sparse priors on $\beta$ and $\sigma^{*2}$. Setting this hyperparameter is straightforward, as we can use a simple line search to find a value that optimally trades off between disregarding the uncertainty estimates (c = 0) and placing too much weights on estimates with low uncertainty (c >= 1). One way to determine the optimal value for c is to randomly permute the biomarker labels, and reduce c until the false positive rate is below some acceptable threshold.

Inference in this model is performed using Hamiltonian Monte Carlo via the Stan software package *Carpenter, 2017*. We report the posterior mode of $\beta$ as well as the posterior probability of observing $\beta > 0$ (if the posterior mode is positive) or $\beta < 0$ (if the posterior mode is negative).

## Acknowledgements

The MJG laboratory is supported by the Wellcome Trust (206194). DW is supported by the NIHR Sheffield Biomedical Research Centre, Rosetrees Trust (ref: A2501), and the Academy of Medical Sciences Springboard (ref: SBF004/1052). MPM is supported by the European Union's Horizon 2020 Research and Innovation Programme (Grant agreement No. 950293 - COMBAT-RES). We thank Benjamin Sidders and Oliver Stegle for feedback on the methodology. We also thank the Sheffield Bioinformatics Core for help with data preprocessing.

## Additional information

### Group author details

**GDSC Screening Team**
Howard Lightfoot; Wanjuan Yang; Maryam Soleimani; Syd Barthorpe; Tatiana Mironenko; Alexandra Beck; Laura Richardson; Ermira Lleshi; James Hall; Charlotte Tolley; William Barendt

### Competing interests

James Hensman: James Hensman is an employee of Amazon.com. The author has no competing financial interests to declare. Jonathan R Dry: Jonathan Dry is affiliated with AstraZeneca and Tempus. The author has no competing financial interests to declare. Frank Dondelinger: Frank Dondelinger is an employee of Roche. The author has no competing financial interests to declare. The other authors declare that no competing interests exist.

### Funding

| Funder | Grant reference number | Author |
| --- | --- | --- |
| NIHR Sheffield Biomedical Research Centre | BRC - IS-BRC-1215-20017 | Dennis Wang |
| Rosetrees Trust | A2501 | Dennis Wang<br>Tzen S Toh |
| Academy of Medical Sciences | SBF004/1052 | Dennis Wang |
| Wellcome Trust | 206194 | Mathew J Garnett |
| Horizon 2020 - Research and Innovation Framework Programme | *Grant agreement No. 950293 - COMBAT-RES* | Michael P Menden |

The funders had no role in study design, data collection and interpretation, or the decision to submit the work for publication.

### Author contributions

Dennis Wang, Conceptualization, Resources, Data curation, Formal analysis, Supervision, Validation, Investigation, Visualization, Methodology, Writing - original draft; James Hensman, Software, Formal analysis, Investigation, Methodology, Writing - review and editing; Ginte Kutkaite, Data curation, Formal analysis, Validation, Visualization, Writing - original draft; Tzen S Toh, Data curation, Formal analysis, Visualization, Writing - original draft, Project administration; Ana Galhoz, Conceptualization, Formal analysis, Supervision, Validation, Writing - review and editing; GDSC Screening Team, Resources, Validation, Data Curation; Jonathan R Dry, Conceptualization, Supervision, Investigation, Writing - review and editing; Julio Saez-Rodriguez, Resources, Data curation, Investigation, Writing - review and editing; Mathew J Garnett, Resources, Data curation, Formal analysis, Investigation, Visualization, Methodology, Writing - original draft, Writing - review and editing; Michael P Menden, Conceptualization, Resources, Data curation, Formal analysis, Supervision, Investigation, Visualization, Methodology, Writing - original draft, Project administration; Frank Dondelinger, Conceptualization, Resources, Data curation, Formal analysis, Supervision, Validation, Visualization, Methodology, Writing - original draft, Project administration, Writing - review and editing

Author ORCIDs
Dennis Wang https://orcid.org/0000-0003-0068-1005
Ginte Kutkaite https://orcid.org/0000-0002-2918-294X
Tzen S Toh https://orcid.org/0000-0003-4211-333X
Ana Galhoz https://orcid.org/0000-0001-7402-5292
Julio Saez-Rodriguez https://orcid.org/0000-0002-8552-8976
Michael P Menden https://orcid.org/0000-0003-0267-5792
Frank Dondelinger https://orcid.org/0000-0003-1816-6300

Decision letter and Author response
Decision letter https://doi.org/10.7554/eLife.60352.sa1
Author response https://doi.org/10.7554/eLife.60352.sa2

## Additional files

### Supplementary files

- Supplementary file 1. Summary of pharmacogenomic associations based on ANOVA.
- Supplementary file 2. Pharmacogenomic associations based on Bayesian testing of GP-curve fits.
- Supplementary file 3. Raw and curve fitted replicate dataset.
- Supplementary file 4. GP-curve fits dataset with calculated summary statistics.
- Transparent reporting form

### Data availability

All data is available through the GDSC downloads portal (ftp://ftp.sanger.ac.uk/pub4/cancerrxgene/releases). Raw dose response data have been deposited in GDSC under v17a_public_raw_data.csv (ftp://ftp.sanger.ac.uk/pub4/cancerrxgene/releases/release-6.0/v17a_public_raw_data.csv). Sigmoid fitted dose-response curves have been deposited in GDSC under v17_fitted_dose_response.csv (ftp://ftp.sanger.ac.uk/pub4/cancerrxgene/releases/release-6.0/v17_fitted_dose_response.xlsx). Cell line genomics data have been deposited in GDSC under GDSCtools_mobems.zip (ftp://ftp.sanger.ac.uk/pub4/cancerrxgene/releases/release-8.0/GDSCtools_mobems.zip). Cell line identity details have been deposited in GDSC under Cell_Lines_Details.xlsx (ftp://ftp.sanger.ac.uk/pub4/cancerrxgene/releases/release-7.0/Cell_Lines_Details.xlsx). Drug compound details have been deposited in GDSC under screened_compunds_rel_8.2.csv (ftp://ftp.sanger.ac.uk/pub4/cancerrxgene/releases/release-6.0/Screened_Compounds.xlsx).

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
