## [Decision Letter]

**Acceptance summary:**

This manuscript presents two statistical approaches to evaluating for drug effect measurements and associations between biomarkers, for dose curve data. Measurements of these kinds are made in many contexts, and frequently reported without accounting well for measurement uncertainties. A statistical framework of this kind will be widely useful and should be frequently applied.

**Decision letter after peer review:**

Thank you for submitting your article "A statistical framework for assessing pharmacological response and biomarkers using uncertainty estimates" for consideration by *eLife*. Your article has been reviewed by two peer reviewers, and the evaluation has been overseen by a Reviewing Editor and Detlef Weigel as the Senior Editor. The following individual involved in review of your submission has agreed to reveal their identity: Alexander W Blocker (Reviewer #1).

The reviewers have discussed the reviews with one another and the Reviewing Editor has drafted this decision to help you prepare a revised submission.

This manuscript presents two statistical approaches to evaluating for drug effect measurements and associations between biomarkers, for dose curve data. Measurements of these kinds are made in many contexts, and frequently reported without accounting well for measurement uncertainties. A statistical framework of this kind will be widely useful and should be frequently applied.

The reviewers noted some issues with in the manuscript, which should be addressed before final acceptance to *eLife*.

Essential revisions:

1) While this work presents a clear statistical framework, the Discussion and Abstract should explain more clearly how this framework improves drug potency or biomarker discovery efforts. That will help communicate the relevance of this work.

2) The authors should clarify of how measurement errors in the curves relate to fitted parameter uncertainties, to explain the superiority of particular fitting approaches.

3) A more thorough discussion of fitting performance is recommended. In particular, accounting for how fitting parameter selection and degrees of freedom. It would also be helpful to explore robust estimators to assess the reliability of fitted parameters. A more thorough analysis is also recommended for edge cases where the IC_50_ falls outside the measured range.

4) Additional model checking and comparisons would be useful. For example, Comparing the presented models to a more standard sigmoid, and additional comparisons highlighted in the comments. It would also help to clarify the relationship between Bayesian posterior probabilities, and ANOVA q-values for the GP approach.

5) The authors should revise Figure 3C and D. Grouping response curves by mechanism can be misleading since different drugs or biomarkers may intrinsically have discordant potencies due to differing on-target binding efficiency or off-target effects.

---

## [Author Response]

Essential revisions:1) While this work presents a clear statistical framework, the Discussion and Abstract should explain more clearly how this framework improves drug potency or biomarker discovery efforts. That will help communicate the relevance of this work.

We have highlighted the utility of our framework for identifying new biomarkers and highlighting drugs with more potency by putting more weight on cases with lower uncertainty in the statistical models as well as generalised to other diseases. We have edited the Abstract:

“High-throughput testing of drugs across molecular-characterised cell lines can identify candidate treatments and discover biomarkers. […] Our method is applicable to any dose-response data without replicates, and improves biomarker discovery for precision medicine.”

and added the following section in the Discussion to communicate this:

“The increasing utilisation of high-throughput drug screening for identifying effective new treatments will necessitate the use of more powerful statistical and machine learning methods (Toh et al., 2019). […] We hope that by considering response uncertainty and providing a probabilistic view of drug biomarkers, the risks associated with drug development can be better balanced and smarter decisions can be made.”

2) The authors should clarify of how measurement errors in the curves relate to fitted parameter uncertainties, to explain the superiority of particular fitting approaches.

First of all, we would like to clarify the term ‘estimation uncertainty’ in the context of our method. This is not related to between-sample dispersion; due to the high-throughput nature of the GDSC assays, no technical or biological replicates were generated, with the exception of the replicate dataset described in Figure 2 for quality control. Hence the estimation uncertainty refers to the standard deviation of summary measures estimated from a single sample. Here, we capture the dispersion of the posterior distribution of these summary measures. If the dispersion is large, then we will have less confidence in the estimated mean value of the summary measure. Measurement errors for individual points on the dose-response curve will generally result in larger estimation uncertainty. We have added a clarification in the Results section:

“When the estimation uncertainty is large, we will have less confidence in the estimated IC_50_ in an experiment. Measurement errors for individual points in a dose-response curve will generally result in larger estimation uncertainty, whereas greater variation between biological replicates will result in larger observation uncertainty.”

Also in Figure 2, we have provided a detailed comparison between this estimation uncertainty and the observation uncertainty that can be measured by estimating summary values for multiple biological replicates. We observe that for the estimation of IC_50_ values, observation uncertainty is strongly positively correlated (r=0.84) with estimation uncertainty when the IC_50_ value is within the dosage range, but for values outside the dosage range, the estimation uncertainty increases, while the observation uncertainty decreases. This indicates that estimation uncertainty is reflective of the true uncertainty in the estimate when the relevant summary statistic (IC_50_) can be determined from the measured dosages alone, but for estimating statistics that require extrapolation beyond the maximum dosage, we would additionally need to take the observation uncertainty into account. We addressed this in the Discussion section by adding this text:

“We are interested in the former to assess the quality of the fit, and therefore the reliability of the estimated IC_50_. […] Instead, our method defines a very large confidence interval for drug response statistics extrapolated beyond the maximum dose tested and we would additionally need to take the observation uncertainty between replicate experiments into account.”

3) A more thorough discussion of fitting performance is recommended. In particular, accounting for how fitting parameter selection and degrees of freedom.

We have added a more detailed discussion and suggestions for selecting GP fitting parameters in the Materials and methods section:

“Hyperparameters for the GP model were determined by manual tuning; however, for other datasets, we could also envision a Bayesian model selection procedure which places the variational inference in a variational-within-MCMC scheme where the MCMC moves update the hyperparameters. […] To avoid massive computational complexity, the MCMC scheme could be run on a representative subsample of cell lines.”

We also added suggestions for how the hyperparameter in the Bayesian biomarker model can be tuned:

“The main tuneable hyperparameter is the scaling parameter c, as the model is robust to changes to the parameters for the sparse priors on *β* and *σ*^*2^. […] One way to determine the optimal value for c is to randomly permute the biomarker labels, and reduce c until the false positive rate is below some acceptable threshold.”

It would also be helpful to explore robust estimators to assess the reliability of fitted parameters.

We agree it would be useful to assess the reliability of the fitted parameters to understand whether the performance of our GP model stemmed from correcting misspecification in original ANOVA, or the use of uncertainty estimates. Like the reviewer, we wondered whether heteroskedasticity of the IC_50_ estimates impacted performance, so we performed our biomarker association testing using the AUCs of the fitted curves, hence AUC values display a homoscedastic behaviour. For this, we show that the uncertainty estimates from the AUCs did not vary across doses like IC_50_ uncertainties (new Figure 4—figure supplement 1A-D). We also show the difference in biomarker associations between sigmoid and GP curve fits (Figure 4—figure supplement 1E, F), and show that we recovered all 26 gold standard biomarkers and additional sensitivity biomarkers using AUCs (Figure 4—figure supplement 1G). These new findings have been added to the Results section:

“Alternatively, AUC values can be used to compare both curve fitting methods (Figure 4—figure supplement 1). While known clinical biomarkers are recovered with AUC as a drug response metric, IC_50_ measures were used in the subsequent analysis as they retain direct relationship with the drug concentration and are more interpretable.”

A more thorough analysis is also recommended for edge cases where the IC_50_ falls outside the measured range.

We have highlighted the analysis we conducted in Figure 2C-E in the Results section:

“Additionally, we examined the relationship between estimation uncertainty and observation uncertainty in a number of edge cases where IC_50_ was estimated within and beyond the maximum concentration tested (Figure 2C-E).”

The reviewer noted that it may not be correct to impose monotonicity when IC_50_ might fall outside the measured range, so we showed that there is much greater IC_50_ uncertainty in those cases. For clarifying this and enhancing the visualisation, we display curve fits within concentration range in solid lines, whilst less confident fits exceeding max concentration are indicated in grey. Furthermore, we have added in red horizontal lines showing the IC_50_ confidence intervals, which are much larger outside of concentration range. We have also provided more rationale and guidance in the Discussion section for interpreting IC_50_s at untested concentrations:

“However, extrapolating beyond the concentration range would be challenging due to the uncertainty in the behaviour of the dose-response curve in unobserved concentrations. […] Instead, our method defines a very large confidence interval for drug response statistics extrapolated beyond the maximum dose tested and we would additionally need to take the observation uncertainty between replicate experiments into account.”

4) Additional model checking and comparisons would be useful. For example, Comparing the presented models to a more standard sigmoid, and additional comparisons highlighted in the comments.

We thank the reviewer for this insightful comment. In order to address this, we have performed a within drug random bootstrap sampling of 80% of all treated cell lines for the Sigmoid model. In particular, we estimated the uncertainty of the model for the drugs dabrafenib and lenalidomide for colorectal and melanoma cell lines, respectively (new Figure 5—figure supplement 1), to compare with GP’s variance estimation (Figure 5B-C). As theorized, the incorporation of uncertainty in the GP model provides a more adequate variance estimation in the prediction of non-responding cell lines IC_50_’s.

An explanation of the methodology was added in the Materials and methods section:

“To estimate the uncertainty of the Sigmoid curve fitting, a random bootstrap sampling of 80% of all treated cell lines available for each drug over 100 iterations was performed. […] The standard deviation of the log(IC_50_) estimates was computed to assess the model’s variance.”

Additionally, a comparison between the models was added in the Results section:

“Additionally, a comparison of the uncertainty estimates for the GP and the Sigmoid curve fitting methods revealed that both display concordant results (Figure 5B and C; Figure 5—figure supplement 1); However, the Sigmoid curve fitting method (Materials and methods; Vis et al., 2013) underestimates variance of non-responding cell lines rendering the GP approach superior.”

It would also help to clarify the relationship between Bayesian posterior probabilities, and ANOVA q-values for the GP approach.

We agree with the reviewer that Bayesian posterior probabilities and ANOVA q-values are different statistical quantities that should not be compared in absolute terms. Posterior probabilities capture the properties of the distribution of a parameter treated as a random value; q-values make a statement about the probability of observing an estimated (fixed) parameter value under the null hypothesis. Nevertheless, we believe that our comparison of these quantities, in relative terms, is valid, because we are interested in the value of using each of them for making a decision about the association between drug sensitivity and biomarkers. ANOVA q-values represent the state of the art, which means that we would expect a broad correlation between the q-values and the Bayesian posterior probabilities, with some deviations in cases where the inclusion of uncertainty estimates helps the Bayesian model detect or discard associations. This is what we see in Figure 5A, which displayed agreement between both statistical quantities and disagreements for a few biomarker associations. The key observation here is that the comparison is in terms of the relative ranking of the Bayesian posterior probabilities and ANOVA q-values, respectively, which enabled us to draw concordant conclusions in terms of prioritising biomarkers of drug response for clinical testing.

We have now added clarifying text in the Results section:

“Since both Bayesian and frequentist methods can be used to prioritise biomarkers for further testing, we compared association statistics (posterior probabilities and q-values) from both statistical methods.”

And the Discussion section:

“While Bayesian posterior probabilities and ANOVA q-values are different statistical quantities for measuring biomarker associations that should not be compared in absolute terms, we compared these quantities in relative terms to prioritise biomarkers of response for further testing.”

5) The authors should revise Figure 3C and D. Grouping response curves by mechanism can be misleading since different drugs or biomarkers may intrinsically have discordant potencies due to differing on-target binding efficiency or off-target effects.

Thank you for touching upon an important point about possible off-targets. While we have information on the primary targets of the drugs, off-targets effects are difficult to measure and define, especially with the histone chromatin targeting drugs we highlighted. Therefore, we classified the target pathways into broad categories that can include many possible target genes (e.g. histone methylation, mitosis). Nevertheless, we have edited the Results and Discussion sections to acknowledge off-target effects as a possible reason for the observed uncertainties. We have also added Figure 3E to show the distribution of uncertainties for individual drugs within the pathways we highlighted and not mislead by grouping drugs. It was reassuring to see that the distributions for individual drugs recapitulate what was seen when we aggregated the drugs into target pathways.